# Identifying the Inertial Properties of a Padel Racket: An Experimental Maneuverability Proposal

**DOI:** 10.3390/s22239266

**Published:** 2022-11-28

**Authors:** Carlos Blanes, Antonio Correcher, Pablo Beltrán, Martin Mellado

**Affiliations:** Instituto de Automática e Informática Industrial, Universitat Politècnica de València, Edificio 8G, Acceso D, 3ª Planta, Camino de Vera s/n, 46022 Valencia, Spain

**Keywords:** padel, moment of inertia, system identification, signal process, racket maneuverability, trifilar pendulum

## Abstract

Although the moment of inertia of padel rackets is one of their fundamental properties and of particular interest to the players, hardly any manufacturer specifies the parameter for its rackets. The present paper offers a solution to determine the moment of inertia around different axes of padel rackets and makes a standardized comparison possible. After a short overview of the physical background of the problem and the existing solutions for inertia testing, the developed concept for a test stand is described in detail. The approach uses the fact that a pendulum swings with its natural frequency, which depends directly on its moment of inertia. The inertia can be calculated by measuring the cycle time of the swing. Two different test stands, a trifilar and a swing pendulum, are designed to enable an oscillation of the rackets with different rotation axes, and an acceleration sensor is used to measure its natural frequency. A user-friendly interface acquires and processes accelerometer data providing inertial moments. A calibration model defines sensor accuracy. Precision is estimated by calculating the influence of the measurement errors and by testing the repeatability. The maneuverability parameter is created, and in the last step, various rackets are evaluated to create a database with the main properties. As a result of the study of the racket population, a maneuverability parameter is proposed to classify the rackets in a comprehensible way for users. The classification method is tested with users to explore the matching between the scientific classification and the player’s feelings. The results are shown and explained.

## 1. Introduction

Padel is a tennis-like racket sport currently gaining popularity across Europe and other continents at an impressive speed. The main differences with tennis are a smaller court (20 × 10 m^2^) with enclosing walls made of glass and fences and solid stringless rackets. Padel is played two against two. The ball has the exact dimensions as the one used for tennis matches. The game was invented in Mexico in 1969, spread during the 1980s in South America, mainly in Argentina, and has been since the 1990s a common and popular sport in Spain with corresponding tournaments, associations, sports clubs with padel courts, and racket manufacturers. Although Spain is the major player country, with over 4 million players [1], padel has also spread outside these countries in recent years [2,3]. Current estimations show that there are over 18 million players and 300,000 federated players around the world. The International Padel Federation (FIP), founded in 1991, is the leading association to regulate and promote padel and is formed by more than 50 national federations representing almost 100 countries [4]. FIP recently launched the Premier Padel^TM^, a global padel tour backed by the Professional Players Association (PPA), with 10 tournaments scheduled for 2022 and 2023 and 25 tournaments for 2024. Another professional circuit, World Padel Tour, was created by a private firm (Setpoint Events, S.A.) that organizes tournaments in cities from more than 15 countries considering Open, Master, and Challenger Tournaments and Exhibitions. The repercussions of padel, the number of players worldwide, and the research interest is increasing [5].

The primary sport implement to play padel is the racket. A padel racket, unlike tennis, badminton, or squash rackets, has no strings; unlike table tennis or pickleball, the padel racket must have at least one hole. There are discussions about the better distribution of the holes in the racket faces. In [6], the flow physics of padel rackets with different hole configurations are studied and tested with the conclusion that the actual design lacks scientific attention and could be optimized with holes in the outer part of the head instead of the center. A padel racket comprises two parts, the head and the handle. The handle must have a maximum length of 20 cm, a maximum width of 50 mm, and a maximum thickness of 50 mm. The length of the head plus the length of the handle may not exceed 45.5 cm, with a maximum width of 26 cm and a maximum thickness of 38 mm. The handle end has a non-elastic cord that must be put around the wrist as protection against accidents [7].

Tennis racket shapes and materials have changed significantly over time, influencing player performance and injury risk [8]. The three main elements of a padel racket are the frame, core, and face. The frame is the component of the racket that surrounds the faces and is usually made in a tubular shape. This is the piece that gives the padel racket strength. That is why high-yield strength is usually used for this component. The frame of padel rackets is typically made of carbon fiber, fiberglass, and aramid fibers (aromatic polyamide).

The core is one of the most critical parts of the racket. It is the material inside, below the face’s surface. The core highly influences the control and power of the racket. Rackets can be divided according to their core material: rubber core as ethylene-vinyl acetate (EVA), foam core, and others, such as polyethylene (PE), or even a hybrid core such as a combination of EVA and PE. In turn, these are divided into several types, depending mainly on their density. For example, EVA rubber can be black EVA rubber, soft EVA rubber, and ultra soft EVA rubber, among others.

Finally, regarding the faces, there is a wide variety of materials for their manufacture. The most common is carbon fiber, which also has subdivisions such as 1K, 3K, 18K, and 24K carbon, depending on its quality and hardness. However, there are also rackets with innovative materials such as graphite or with more ordinary materials such as fiberglass.

Because of the growth of padel sport, scientific research and padel racket sales have increased notably in the last few years [9,10]. That commercial opportunity has led to many firms dedicated to padel racket manufacturing. There is a wide variety of padel rackets on the market, referring to many brands, models, colors, and different materials for their manufacture. Padel rackets are handcrafted, and there is no standard for this process. The manufacturing process is mainly artisanal, with different stages from design to molding, deburring, forming, priming, painting, hole drilling, and finishing. The components and materials with which the padel rackets are made influence their durability, performance, type of play, and bending stiffness [11].

When a company develops a new racket model, it defines its characteristics based on its experience and commercial interests. This lack of standardization is a problem for users, who cannot compare rackets with each other except by direct testing, and for companies, as it is complex to determine the characteristics of a particular racket design beyond evaluation by expert users. This problem is currently being focused on by researchers in disciplines ranging from rackets materials technology [12] to creating an intelligent tennis coach for a racket [13] or in table tennis [14] and with wearable sensing devices for whichever racket sports training [15], which is of interest to both companies and padel players, as well as to researchers. In order to measure the player experience, the leading tennis racket firms have developed wearables to characterize some game characteristics. The Sony Smart Tennis Sensor (Sonny, Weybridge, UK) and the Head Tennis Sensor (HEAD, Phoenix, AZ, USA) are examples of these devices. They are attached to most tennis rackets’ bottoms and measure the kind of stroke (righthand, lefthand, smash, volley, sliced, or topspin). They also give the impact point and the ball’s speed. Another wearable with similar characteristics is the Zepp Tennis 2 (Zepp Labs Inc., San Hose, CA, USA) but it offers a flex mounting configuration that adapts to any racket. These devices offer video analysis tools in their apps to analyze the player’s strokes.

This is the goal of the Laboratory *Testea Padel* at Universitat Politècnica de València, the first and only laboratory in the world entirely dedicated to researching padel rackets. The research in this laboratory is oriented to allow the characterization of padel rackets to know all their parameters related to mechanical properties and playable characteristics. During the last five years, several test stands have been designed to determine different parameters of the padel rackets. One of the playable characteristics analyzed for padel rackets is maneuverability, that is, the quality of being easy to displace and rotate. The maneuverability of a padel racket is an essential feature in specifying its playable level. It depends on the racket weight, the location of its center of mass (COM), and its mass distribution, that is, its moment of inertia (MOI).

Although the MOI of padel rackets is one of their fundamental properties and is of particular interest to tennis players [16,17], and MOI is more critical than racket mass in the swing speed of the racket [18], hardly any manufacturer specifies this parameter for its rackets. MOI is also an essential feature in other sports such as golf [19] and cricket [20]. In the case of tennis, the effects of the racket’s physics on the player’s performance are already well studied by researchers for decades. One of the most important characteristics of the rackets is the moments of inertia around the axes through the handle. Various works [21,22,23,24,25] investigated the MOIs for tennis rackets and their effects on the player’s experience and the risk of injuries in the elbow and wrist when players served [24]. In order to measure the tennis racket MOIs, there are mathematical [21] or experimental proposals [26,27] (Spurr, 2016) (Allen, 2018). Nevertheless, work regarding MOI measuring or estimation has yet to be found for padel rackets.

The aim of this work is to design a test stand to enable the evaluation of MOIs for padel rackets. Figure 1 shows the axes of rotation that interest this study.

The MOI around the horizontal axes (here, x and y axis) is commonly known as the swing weight and measures how “heavy” the racket feels when it is swung. The study in [17] about tennis rackets with different participants points out that “swing speed decreased with respect to the MOI according to a power relationship.” The same result can be found in [28,29]. In the case of the vertical axis (z-axis), the MOI determines the stability of the racket when the ball hits the racket off-center. According to [23], higher inertia increases the racket’s resistance to twisting. The same applies to padel rackets, which shows that the analysis and measurement of their inertial properties is an essential and present issue for manufacturers and players.

As padel is a novel sport, there is still not much research on rackets’ properties. This paper proposes a methodology to measure their maneuverability and a test stand to make measuring tests and present results on padel rackets. Therefore, the present paper offers a solution to determine the MOI around different axes of padel rackets to make a standardized comparison possible. Once the test stand is calibrated, a population of 137 padel rackets is measured. The mass, balance, and MOIs around the x, y, and z axes are measured for each racket. With these data, a maneuverability index combining the inertial properties of the rackets is proposed. This parameter allows the classification of the maneuverability of a racket in a comprehensible way for a user. A study with 50 padel users is presented to verify the matching between the scientific index and the player’s feelings.

## 2. Measurement Process

### 2.1. Preliminary Measurements

The weight of the padel racket is the first parameter with more influence on maneuverability. A heavy padel racket is harder to move. The padel racket’s COM corresponds with its balance point, where it can be balanced without tipping over. A simple gadget can quickly determine this point (Figure 2). The padel racket is positioned on a rotatable metal bar, and with the attached knob, it is possible to rotate the bar until the racket surpasses the balance point and tips over. The distance between the handle end and the COM, the balance distance, can then be easily read from the ruler on the table.

From our study of 137 padel rackets (Figure 3), the minimum and maximum values for the weight of padel rackets are 0.316 kg and 0.387 kg, with a mean of 0.363 kg. The balance of the padel rackets is in the interval of 246 mm to 283 mm, with a mean of 265.52 mm.

### 2.2. MOI Calculation Methods

Several methods experimentally can be used to determine the moments of inertia of irregularly shaped bodies. The Atwood machine was invented in 1784 by the English mathematician George Atwood as a laboratory experiment to verify the mechanical laws of motion with constant acceleration. The Atwood machine is an easy and familiar way to demonstrate the principles of inertia and acceleration and can also be used to determine the MOI of different bodies. Even if this method has the advantage of the easy calculation of inertia moments, the experiments are too inconvenient, as it takes a long time to wind up the thread every time to prepare the setup, and it requires a controllable motor.

Another way to determine the inertia of a given object is to let it oscillate freely like a physical pendulum around a point out of the COM of the object. When a pendulum is neither driven nor damped, it oscillates with its natural frequency, which depends directly on the MOI. Thus, the time of one oscillation period can use to calculate the inertia. In the case of a rotation around a horizontal axis, the motion of the swinging object can be described with the already derived equation for physical pendulums:(1)$$-mgd\sin\theta =I\ddot{\theta },$$
where *θ* is the rotation angle, *I* is MOI through the rotation pivot for an object, *m* mass, *d* distance between the rotation pivot and the COM and, with *T* the time of period, it is possible to calculate the MOI *I* according to:(2)$$I=\frac{T^{2}mgd}{4\pi^{2}},$$

The cycle time of one oscillation can be measured with a slow-motion camera, encoder, or accelerometer sensor. To determine the distance *d,* the COM must be located before. Since it coincides with the point where the object can balance without tipping over, it can be found by balancing the object and considering the symmetry. The application of this pendulum experiment to tennis rackets is well known, as it can be found in [22], amongst others.

A third method to compute the inertia of an object is the oscillation around the vertical rotation axis. An easy way to enable a free vertical oscillation is to suspend the respective object on one or more threads. A torsional pendulum with three threads or cables is called a trifilar pendulum [30]. The strings are arranged in a triangle with a constant distance *R* to the axis of rotation (Figure 4).

Adding wires to the basic torsional pendulum improves the setup’s stability, and the added threads’ torsional stiffness is insignificant. The pendulum hangs from wires of identical length and rotates vertically around its COM. Friction and mass of the threads are neglected, and the weight force of the rotating body is divided uniformly between them. In addition, small angular displacement for the angles (Figure 5) assumes that *tanθ ≈ θ* and *tanα ≈ α* are valid. These assumptions are ordinary simplifications found in different authors [31,32].

Because of the enhanced stability, trifilar pendulums with three threads are used in most applications. With this setup, the time period depends again directly on the MOI. The parameters needed to calculate the inertia around a vertical axis are the length *L* of the threads, the distance *R* between the threads and rotation axis, the mass *m*, and the period *T* of oscillation. After measuring these parameters, the pendulum MOIs, *I*, can be calculated according to Equation (3).
(3)$$I=\frac{T^{2}R^{2}mg}{4\pi^{2}L}$$

The pendulum mass has two components, the mass of the plate *m_p_* and the mass of the racket *m_r_* (Equation (4)), and knowing the pendulum base plate MOIs, *I_p_*, it is possible to obtain the racket MOI *I_r_*.
(4)$$m=m_{p}+m_{r}$$
(5)$$I=I_{p}+I_{r}$$

For example, there are many applications of trifilar pendulums to evaluate rotors [33] and airplanes [32]. The application of this experiment and its systematic measurement errors are well studied. In [31,34], they specify that the COM of the rotating body must pass through the rotation axis to obtain exact results. A solution for this problem is proposed in [30], using load sensors and additional weights on the lower disk to adjust the COM, and in [35], using a universal joint to align de COM.

MOI can also be estimated with torsional springs. The racket is placed on a rotating disc, then displaced in an angular direction, the spring tension exerts a torque on the disc, and it starts to oscillate. Hooke’s law determines the torque applied. The equation of the MOI is the same as for the torsional pendulum with one string. The object’s MOI is calculated if the spring constant and MOI of the platform disc are known. With an alternative design, it is also possible to use linear springs to stimulate the oscillation [36].

### 2.3. Comparison of the Approaches

Every one of the presented solutions has advantages and disadvantages to consider. Table 1 shows the main issues of the approaches.

The Atwood machine needs a long time to wind up the thread every time to prepare the setup and the complexity of this device is higher because a controllable motor is recommended, if not mandatory. The approaches with the pendulum and the springs are similar, but the pendulum solution is selected more frequently. Developing a device with springs is more complex than a pendulum because the spring Young’s modulus should be known. Moreover, the approach with the pendulum yields good results in other studies for tennis rackets [21,27].

### 2.4. Test Stand Design and Implementation

The following section describes the mechanical designs of the test stand to enable an oscillation of padel rackets. Padel rackets have three MOIs axes of interest (Figure 1). Two designs are needed to measure the inertia around the horizontal axes (x and y) and the vertical axis (z). The first is a trifilar pendulum used for the MOI of the vertical axis z. This method has higher stability than the simple torsional spring pendulum and no Young’s torsion modulus needs to be determined. The second is a horizontal rotation pendulum used for the MOI of axes x and y.

The rackets in the horizontal pendulum are assembled between two 3D-printed parts and pivot around a horizontal axis (Figure 6). The limiter part is fixed with a blinder clip to facilitate the adjustment of the racket paddle in the correct position (with the end of the handle 80 mm from the rotation axis) before holding it with screws. The padel racket cradle and holder have a V-shape identical to the racket handles. The cradle has a shaft that pivots around two ball bearings to enable oscillation. With this design, COM of the padel rackets lies right under the rotation axis. A stopper limits the racket positions that rotate around an axis that passes 8 cm from its bottom. The chosen distance in other studies for tennis rackets ranges from 7 cm [28] through 7.6 cm [27] to 4 inches (10.2 cm) [17]. The present work sets the distance to the mean value of 8 cm. This distance is adequate to represent the area where the player holds the padel racket.

A trifilar pendulum is used to determine the MOI around the vertical axis. The rotation body is hung up with three strings to enable oscillation (Figure 7).

The padel rackets are mounted between two polylactic acid (PLA) 3D-printed components; the cradle and the holder are shaped like the handle of rackets.

The rotating base is hanging with three parallel strings. The strings are fixed with nodes and keep parallel the upper and the rotating base. COM must lie on the rotation axis, so the holders are designed so that the padel racket’s vertical axis z coincides with the rotation axis. The COM of the cradle and holder with fasteners lies not precisely on this axis. The MOI calculated with CAD software SolidWorks^®^ (Dassault Systèmes, France, 2022) of the test stand is 2.16181 kg cm^2^, higher than the MOI that passes through the COM 2.157 kg cm^2^. That does not significantly influence the results since the vertical MOI of padel rackets is higher.

The thread length *L* must be measured. The effective length depends on the actual angular position of the lower holder. The string length is *L* = *d_1_* + *d_2_* + *d_3_* (Figure 7) when the holder is not displaced because the thread knots are on the holders’ outer sides. After displacing the holders with the racket, the effective thread length is *d_2_*. To consider the observation, the mean value *L* = *d_2_* + (*d_1_* + *d_3_*)/2 is used in the follow-up experiments and calculations.

Previous prototypes determine the final design. This test stand has different free oscillation modes; vertical rotation and swinging displacements in horizontal directions. The length of the three strings *L* and its distance *R* determines the frequency of those modes. The vertical rotation mode determines the z axis padel racket MOI. This length was chosen to separate these frequencies. Short string lengths and radii make a swinging displacement in the horizontal direction easier. The final compromise solution was string length *L* = 150 mm, and *R* = 100 mm radius distance.

### 2.5. Sensor Data Processing

A biaxial accelerometer is used to determine the cycle time of the oscillation. The accelerometer signal’s frequency coincides with the vertical oscillation movement frequency. The signal’s amplitude does not matter, so the position of the sensor is not significant. As shown in Figure 8, the sensor is placed as far away from the rotation axis as possible to obtain a high amplitude and, thus, a less noisy signal. For the case of the vertical pendulum, it is attached to the plate, and the horizontal pendulum is on the racket itself since the holders hardly move.

The used sensor is an ADXL Dual-Axis Accelerometer with ±5 g full-scale range [37] from the manufacturer Analog Devices. Its data are acquired through the National Instruments I/O device USB-6008 [38] with a sample rate of 1 kHz. Figure 9 shows the experimental setup block diagram.

A MATLAB^®^ function process the data and determine the period time of the oscillation. It acquires the raw data from the I/O device for a determined time, optimizes the signal in several steps, and performs a fast Fourier transformation (FFT) to obtain the power spectral density. The first and higher harmonic is the vertical oscillation frequency. The different steps of the processing signal are:Raw sensor data are noisy and coarse, so signals are filtered first with low pass at 1500 Hz and second with a simple moving average. It replaces every data point by the mean of its initial value and the value of the preceding and the following two data points. Filters reduce the noise, and the leaps in the signal are smoothed out, as shown in Figure 10.The signal is trimmed (Figure 10b) to consider only a whole multiple of the repetitive period. That means that the dataset starts at the same point of the period where it ends. An offset is applied to the data so that the mean of the values is just zero. After that, complete cycles can be detected by searching for the points where the sign changes from positive to negative or vice versa.It shows the frequency spectra (Figure 11). After obtaining the dominating frequency, the period determines the time of an entire cycle.

### 2.6. Interface Design Software

The developed MATLAB^®^ interface facilitates the experiments with the test stand. Apart from the primary function of data acquisition and calculating the MOI from the sensor data and racket parameters, it visualizes the raw (Figure 10a), optimized signal (Figure 10b), and the computed period (Figure 11). Figure 12 shows a screenshot of the interface. Every pendulum needs different parameters (Equations (2) and (3)). This selection enables and disables the corresponding edit fields for the parameters. Before measuring, the I/O device is connected by specifying the device’s name. The measurement starts with an adjustable sample frequency and total sample time. After measuring, the program displays the original sensor signal, the optimized signal, and the resulting period duration from the FFT. The frequency spectrum can be plotted additionally.

## 3. Calibration, Testing, and Error Estimation

This section estimates the accuracy of measuring using objects with simple geometry. The inertia of these objects can be calculated easily. The theoretical values can be compared with the measured ones to verify the results.

### 3.1. Propagation of Error and Repeatability for Horizontal Rotation

The first approach to estimate the accuracy is to identify the sources of measurement errors and calculate their influence on the results. The propagation error approach calculates the effect of every measurement parameter. Since the parameters for the horizontal and vertical rotation axes are different, their error propagation is treated separately. The whole experiment is repeated several times for the same padel racket to determine how much the results diverge in every repetition.

Equation (6) defines the moments of inertia of the racket *I_r_* around the horizontal rotation axes x and y.
(6)$$I_{r}=\frac{gT^{2}\left(m_{0}+m_{r}\right)d}{4\pi^{2}}-I_{0}$$
where *m*_0_ is the mass of the test stand without the padel racket, *m_r_* is the mass of the padel racket, *I*_0_ is the MOI with no load, *T* is the period, and *d* is the distance between the rotation pivot and the COM. The cradle, holder, screws, and nuts are set to have an MOI *I*_0_ that must be subtracted from the result. The COM of the test stand without a padel racket lies on the rotation axis. Hence, it cannot oscillate without a racket and must be estimated using CAD software SolidWorks^®^. The approximation cannot be very accurate because the software assumes uniform density for every part, which is not the case for 3D-printing parts. MOI of the empty test stand is 24.998 kg cm^2^. Nevertheless, the approximation shows that *I*_0_ is a mere fraction of the MOI of a padel racket (*I*_0_/*I_r_* ≈ 0.1%), so the influence of the errors in *I*_0_ is insignificant.

The distance *d* between the rotation axis and COM of the complete setup with a racket depends on their separate COMs (Equation (7))
(7)$$d=\frac{d_{0}m_{0}+d_{r}m_{r}}{m_{0}+m_{r}}$$
where *d*_0_ and *d_r_* are the distances between the rotation pivot and the COM of the non-loaded stand and the racket respectively. As for this case, *d*_0_ = 0 applies because of the test stand symmetry, thus Equation (6) simplifies to:(8)$$I_{r}=\frac{gT^{2}m_{r}d_{r}}{4\pi^{2}}-I_{0}$$

That means that measurement errors of the period time *T*, the mass *m_r_*, and the distance *d_r_* cause deviations in the final results for the MOI *I_r_*. According to error propagation, the standard deviations Δ*T*, Δ*m_r_*, Δ*d_r_*, and the partial derivatives to the parameters, must be determined to calculate Δ*I_r_*.

The standard deviation of every parameter involved in MOIs (Equation (8)) is calculated by measuring the same padel racket with the devices developed ten times.
(9)$$\frac{\partial I}{\partial T}=\frac{gTm_{r}d_{r}}{2\pi^{2}}\;\;\;\;\;\;\;\;\;\;\;\;\;∆T=\pm 1.21\;\mathrm{ms}$$
(10)$$\frac{\partial I}{\partial m_{r}}=\frac{gT^{2}d_{r}}{4\pi^{2}}\;\;\;\;\;\;\;\;\;\;\;\;\;∆m_{r}=\pm 0.516\;g$$
(11)$$\frac{\partial I}{\partial d_{r}}=\frac{gT^{2}m_{r}}{4\pi^{2}}\;\;\;\;\;\;\;\;\;\;\;\;\;∆d_{r}=\pm 0.425\;\mathrm{mm}$$

To compare the magnitude of the parameter’s influences in error, the average values of the N padel rackets are *T* = 1.103 s, *d* = 0.026552 m, and *m* = 0.36307 kg. These values are used as references. Table 2 shows the analyzed N padel racket’s average values, partial derivations, and standard deviation. The multiplication of standard deviations and partial derivations are used to compare their influence on the final result. The measurement error of *d_r_*, *m_r_*, and T has a similar influence.

According to error propagation, considering the resulting maximum error of 10% for *I*_0_ from estimated with the CAD, the standard deviation can be estimated Δ*I*_0_ could have a maximum value of 0.025 kg cm^2^. Concerning the calculated average value of 137 padel racket, MOI in axes x and y *I_r_* = 1.963 × 10^−4^ kg m^2^, the error propagation amounts to 0.0127%.

The uncertainty or error propagation (*µ_xy_*) of the calculation of *I_r_* from horizontal pendulum rotation is shown in Equation (12) and is 0.820 kg cm^2^, where standard deviations were taken to be error estimation. The error propagation amount is 0.418%, considering average values of MOI x-y 196.313 kg cm^2^.
(12)$$\mu_{xy}=\sqrt{{\left(\frac{\partial I}{\partial T}.∆T\right)}^{2}+{\left(\frac{\partial I}{\partial m_{r}}.∆m_{r}\right)}^{2}+{\left(\frac{\partial I}{\partial d_{r}}.∆d_{r}\right)}^{2}}\;$$

Another way to estimate the accuracy of the test stand is to determine the deviation between the results MOI of different experiments with three different references. References were selected to represent the lowest, average, and highest MOI analyzed from 137 padel rackets. The models (a, b, and c) used are pine wood bars which have been measured 10 times and average values are: diameter (0.035, 0.0347, and 0.0345 m), length (0.575, 0.547, and 0.532 m) and mass (0.2899, 0.2977, and 0.2944 kg) (Figure 13). 

Those data provide the theoretical MOI that is compared with the measured in Table 3. Results show an accuracy error average of 0.490% and a repeatability average of 0.215%, considered as the coefficient of a standard deviation compared to the theoretical value.

### 3.2. Propagation of Error and Repeatability for Vertical Rotation

For the vertical rotation axis, the equation to calculate the racket’s MOI is:(13)$$I_{r}=\frac{gR^{2}T^{2}\left(m_{0}+m_{r}\right)}{4\pi^{2}L}-I_{0}$$

The inertia of the vertical test stand without racket *I*_0_ must be subtracted from the result to obtain the pure inertia of the padel racket. It is determined by measuring the time of the period without racket *T*_0_. The equation is:(14)$$I_{r}=\frac{gR^{2}}{4\pi^{2}L}\left(T^{2}\left(m_{0}+m_{r}\right)-T_{0}^{2}m_{0}\right)$$

Table 4 shows the results of the measured parameters and their corresponding partial derivatives from the equations:(15)$$\frac{\partial I}{\partial T}=\frac{gTR^{2}}{2\pi^{2}L}\left(m_{0}+m_{r}\right)\;\;\;\;\;\;\;\;\;\;\;\;\;∆T=\pm 1.08\;\mathrm{ms}$$
(16)$$\frac{\partial I}{\partial T_{0}}=-\frac{gT_{0}m_{0}R^{2}}{2\pi^{2}L}\;\;\;\;\;\;\;\;\;\;\;\;\;∆T_{0}=\pm 1.6\;\mathrm{ms}$$
(17)$$\frac{\partial I}{\partial m_{r}}=\frac{gT^{2}R^{2}}{4\pi^{2}L}\;\;\;\;\;\;\;\;\;\;\;\;\;∆m_{r}=\pm 0.5164\;g$$
(18)$$\frac{\partial I}{\partial m_{0}}=\frac{gR^{2}}{4\pi^{2}L}\left(T^{2}-T_{0}^{2}\right)\;\;\;\;\;\;\;\;\;\;\;\;\;∆m_{0}=\pm 0.1678\;g$$
(19)$$\frac{\partial I}{\partial L}=-\frac{gR^{2}}{4\pi^{2}L^{2}}\left(T^{2}\left(m_{0}+m_{r}\right)-T_{0}^{2}m_{0}\right)\;\;\;\;\;\;\;\;\;\;\;\;\;∆L=\pm 0.0868\;\mathrm{mm}$$
(20)$$\frac{\partial I}{\partial R}=\frac{gR}{2\pi^{2}L}\left(T^{2}\left(m_{0}+m_{r}\right)-T_{0}^{2}m_{0}\right)\;\;\;\;\;\;\;\;\;\;\;\;\;∆R=\pm 0.1714\;\mathrm{mm}$$

The uncertainty or error propagation (*µ_z_*) of the calculation of *I_z_* from vertical trifilar pendulum rotation is shown in Equation (21) and is 0.114 kg cm^2^, where standard deviations were taken to be error estimation. The error propagation amount is 0.658%, considering the average values of MOI axis z 17.282 kg cm^2^. This result is consistent with the findings [39] in the error analysis with tennis rackets, where period T and plate radius R are the most influential parameters in the relative error. According to Equations (15)–(20), if the length of the threads increases, the relative errors decrease.
(21)$$\mu_{z}=\sqrt{{\left(\frac{\partial I}{\partial T}.∆T\right)}^{2}+{\left(\frac{\partial I}{\partial T_{0}}.∆T_{0}\right)}^{2}+{\left(\frac{\partial I}{\partial m_{r}}.∆m_{r}\right)}^{2}+{\left(\frac{\partial I}{\partial m_{0}}.∆m_{0}\right)}^{2}+{\left(\frac{\partial I}{\partial L}.∆L\right)}^{2}+{\left(\frac{\partial I}{\partial R}.∆R\right)}^{2}}\;$$

In the same way, horizontal pendulum accuracy has been determined with three different references. References were selected to represent the lowest, average, and highest MOI analyzed from 137 padel rackets. The models (A, B, and C) used are aluminum bar profiles (0.045 × 0.045 m^2^) which have been measured 10 times and average values are: length (0.247, 0.236, and 0.225 m) and mass (0.381, 0.367, and 0.347 kg) (Figure 14).

Those data provide the theoretical vertical MOI compared with the measured ones in Table 5. Results show an accuracy error average of 0.990% and a repeatability average of 0.303%, considered as the coefficient of a standard deviation compared to the theoretical value.

### 3.3. Summary on Error Estimation and Considerations

Table 6 summarizes all results for the relative deviation from this section. Since the error propagation is rather theoretical and depends heavily on the estimated measurement deviations, the repeatability and the comparison to theoretical values provide more precise information about the accuracy of the test stand.

The vertical test stand has a 0.5% of error estimation and the horizontal one 1%. Therefore, the experiments with the test stand yield results with a reasonable maximum deviation of approximately 1% and high repeatability. The two setups for the different orientations have similar repeatability and even more because the error propagations have been estimated with a confidence level of 68% in a normal distribution, correspondence to a one-sigma effect.

Apart from the mentioned measurement errors, some other aspects affect the experiments. During the testing of the padel rackets, the following issues were found:Some rackets are manipulated with extra grips or protectors on the side. These items change the weight and balance point.The variance of the parameters between rackets of the same model is significant. Some manufacturers specify the weight of the rackets with a range of 20 g, which causes a variance that makes it impossible to generalize the results for all rackets of the same model accurately.The handle end of some models is oddly shaped or has a strap attached, making it difficult to accurately measure the distance to the COM and position the racket in the cradle.Even though ball bearings used in the horizontal pendulum have low friction, this friction damping the swing, and, after approximately ten cycles, the padel rackets stop due to the friction force.In the case of the vertical rotation axis, the rotation goes not precisely through the COM; since the thickness of the padel rackets varies, the axes do not always coincide.The trifilar vertical pendulum achieves worse results than the horizontal pendulum. However, it allows calculating the MOI when the pendulum’s axis passes through the COM of the padel racket, as occurs with the z axis. The trifilar pendulum requires a more complex and heavier cradle to fix the padel racket, the slight deviations between the pendulum axis and the distance to the COM influence the result. The initial manual impulse given to the trifilar pendulum makes it not only oscillate around its vertical axis, creating an overlapping of harmonic movements with different frequencies that add error to the measurement. The horizontal pendulum is an effortless and reliable design, but it only can be used when the axis of rotation of the pendulum and the COM are significantly separated.

## 4. Experiments and Results

### 4.1. Correlation between Values

To put the test stand into operation and to test the developed interface, a complete evaluation of all available padel rackets is realized. A database with 137 different padel rackets was created. The results are analyzed in this section to find correlations between the measured values.

For every racket, the mass *m*, the distance *d_com_* between handle end and COM (i.e., balance distance), and the moments of inertia *I_x_*, *I_y_*, *I_z_* through the COM are measured. Figure 1 shows the corresponding axes. Table 7 summarizes the data measured.

Results show no wide range values in any parameter because padel rackets have size requirements to be accepted as good rackets in FIP [7].

The MOI measurement is repeated three times for every racket and orientation, and the average value is calculated. As expected, a strong correlation between the balance distance and the MOIs in x and y axes can be found. Figure 15 shows that the further the balance point is located from the handle end, the higher the inertia in both directions.

Moreover, from the graph form, the correlation can be expressed by the Pearson correlation coefficient, which measures linear correlation and amounts to 0 for no linear correlation and −1 or 1 for a total linear correlation. The result for the coefficient between *d_com_* and *I_x_* is 0.82, and for *d_com_* and *I_y_* is 0.80. In addition, the two considered moments of inertia correlate strongly with each other. As shown in Figure 15, a padel racket with a high result for *I_x_* also has a high result for *d_com_* (balance) and also a high result for *I_y_ (*Figure 16). Here the determination correlation coefficient (R^2^) amounts to 0.88 and 0.97, removing two outsiders.

These findings can be used to interpret future measurements. When the result of an experiment with standard rackets differs substantially from the values in the database, it indicates a measurement error.

Considering the MOI around the z axis, *I_z_*, no correlation can be found. As expected, the location of the COM does not influence the inertia around the z axis. Moreover, neither the mass nor other axis MOI shows any correlation. The highest correlation coefficient amounts to only 0.41 between *I_y_* and *I_z_*.

### 4.2. Maneuverability Parameter

The maneuverability of a padel racket can be defined as the ability to be moved with more or less difficulty (the feeling of heaviness or lightness of a racket moving). Maneuverability is one of the most important aspects for a beginner or intermediate player and, generally, for players with limited muscle strength. As described in the introduction section, maneuverability depends on the racket’s weight, the COM’s location, and the weight distribution. Those characteristics are represented by the moments of inertia. The higher the MOI, the lower the maneuverability because the user will have to generate higher torques to produce the same acceleration.

The main intention of the experiments is to determine the mechanical properties of every padel racket and determine a specific parameter of the padel rackets that is easily understandable for the players and the manufacturers as well as valuable to compare different padel rackets. Since the specification of too many values overwhelms the customer without helping to find a suitable racket, the specification should be minimal and without redundant information. For example, the actual values of the MOIs are not easy to understand, as there are three possible values for x, y, and z axes. In addition, the units used (kg cm^2^) are not comfortable outside of technical language. 

We propose a normalization of the MOIs from a reference padel racket to make the parameters comparable. We chose the racket with the lower distance to the average of each MOI as a reference. Therefore, a normalized MOI with a value of “1” means a maneuverability response similar to the pattern. Values larger than “1” mean less maneuverability, and under “1” mean more maneuverability. Each one of the normalized MOIs represents the maneuverability related to its axis. Therefore, we propose the average of the normalized MOIs for each axis to have a global maneuverability performance index (*I_p_*).
(22)Ip=IxIxR+IyIyR+IzIzR3
where *I_x_*, *I_y_*, and *I_z_* are the MOIs of the racket on each axis and *I_xR_*, *I_yR_*, and *I_zR_* are the MOIs of the reference racket on each axis.

Figure 17 shows the results obtained for the studied padel rackets.

Rackets are grouped with each 10% index increment, which means that it takes 10% more effort (10% higher torque) for the player to generate the same acceleration with a racket belonging from one group to the next. With the proposed index, all the rackets in this study can be grouped into three categories. Most rackets belong to the central category, as expected, because manufacturers try to design rackets that can be used for most padel player levels. Higher maneuverability involves less racket weight, which is a good feature for executing control strokes (low-speed strokes). Nevertheless, power strokes are easily executed with higher racket weights (lower maneuverability). Therefore, rackets with medium maneuverability are selected to have both control and power characteristics in the racket.

## 5. Study with Padel Users

One of the goals of this work and the Testea Padel Laboratory at Universitat Politècnica de València is to provide comparable and comprehensible parameters of the tested padel rackets for players and manufacturers. In this section, padel players looking for a racket to buy are asked to estimate the maneuverability of some selected padel rackets. The results are then compared to the calculated values of the maneuverability index in Section 4 to assess if the index is aligned with the user sensations.

### 5.1. The Procedure of the Study

The users must order three padel rackets according to their maneuverability (high, medium, and low). They are told to rotate the rackets around every axis and then move the racket freely to feel which one seems to be easy to move. Rackets with indexes 0.90, 1.02, and 1.12 were selected to facilitate the classification.

### 5.2. Results of the Study

The study involved 50 users. Table 8 and Figure 18 show the results of the test. 

The results show that most users detect the padel rackets with the highest and lowest maneuverability index. Data also show that medium and low maneuverability rackets are more challenging to classify. Nevertheless, most users feel less maneuverable rackets with higher index values. 

Although this study lacks more testing with more samples, the results suggest that the maneuverability index ranges correspond with the impression of the players.

## 6. Conclusions and Future Work

This work shows the complete development of a test stand to determine the moments of inertia of padel rackets around different axes. The rackets are mounted to oscillate freely like a pendulum to calculate the inertia. The moment of inertia is calculated by determining the time of one oscillation as well as the racket’s weight and balance point.

Two different designs for mounting the rackets are developed to enable horizontal (axes x and y, Figure 1) and vertical (axis z, Figure 1) oscillation. The cycle time of oscillations is measured with an accelerometer, and its signal is processed with the fast Fourier transformation. The cycle time can be calculated by searching for the dominant frequency in the signal. Software with a graphical user interface is programmed to perform the experiments efficiently. The sensor data are acquired, visualized, and processed directly in the application, and the calculated results are stored in a database.

After the set-up of the test stand, the results are verified, and the accuracy is estimated. Their moment of inertia can be calculated easily with basic Equations (8) and (14). Comparing the values yields an accuracy error between 0.49% for the horizontal test stand and 0.99% for the vertical test stand. The measurement errors are in line with the error propagation calculated. The result is a maximum deviation of 1.71% for the worst case. Moreover, the repeatability is determined by evaluating the same racket several times to show that the results are stable and do not differ between experiments with the same padel racket. The relative standard deviation between the references’ results amounts between 0.215 to 0.303%.

The vertical trifilar pendulum has less accuracy and repeatability than the horizontal pendulum, and this could be because, in the vertical pendulum, oscillations are not only vertical. Other oscillations, belonging to their secondary natural oscillation modes, are added to the vertical one. The calculated values of the periods have a higher relative standard deviation (1.08 ms vs. 1.21 ms) despite the vertical period average being two times less than the horizontal pendulum. 

Finally, 137 different padel rackets are evaluated with the test stand to create a database and analyze the results. As expected, the MOIs in x and y axes correlate strongly with the distance between the rotation axis and the center of mass. 

In addition, a maneuverability index to classify the rackets is proposed. This index can be used to compare rackets from different brands and provide comprehensible information to the user to help him choose a padel racket. Experiments with padel users showed that their racket evaluation aligns with the test stand results and the classification proposal. The lower moment of inertia around the three axes the rackets have, the better feels the maneuverability for the players. Future work could explore the generation of a software tool to help users choose a racket depending on their preferences. Maneuverability and other aspects such as price, materials, design, or shape should also be included.

Since the work is part of the padel racket evaluation of the Testea Padel Laboratory at Universitat Politècnica de València, the test stand will be used for further racket testing in the future. Then, it will be possible to add improvements or new features to the interface in case issues or new ideas arise during usage. Further studies with expert players can be conducted to learn more about how the different parameters influence the playing experience and how manufacturers can specify them efficiently to help players find the most suitable racket without overwhelming them with numbers. With the current solution and design, there is no easy way to increase the accuracy further. If more accuracy is required, another test stand with one of the other approaches can be developed to compare the precision of the results.

## Figures and Tables

**Figure 1 sensors-22-09266-f001:**
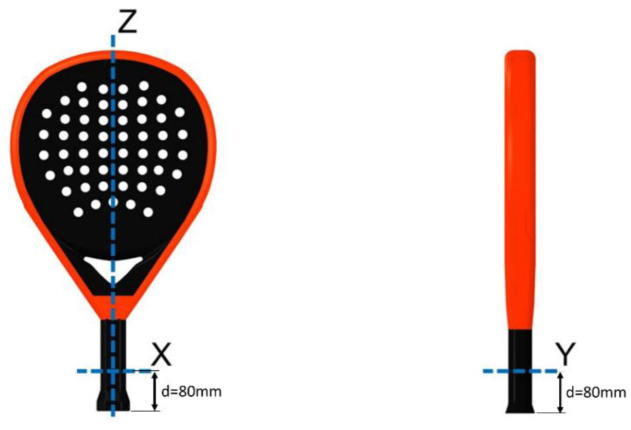
Padel racket and its rotation axes.

**Figure 2 sensors-22-09266-f002:**
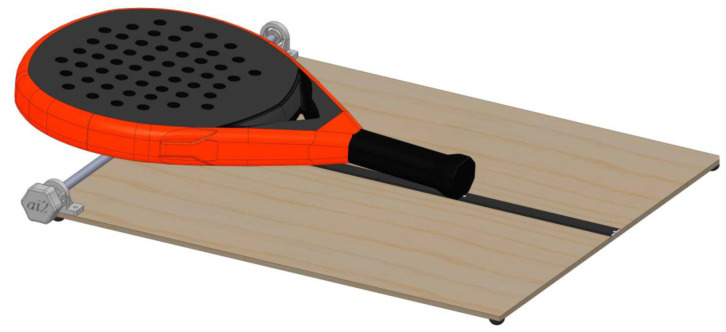
The device used to determine the COM of padel rackets.

**Figure 3 sensors-22-09266-f003:**
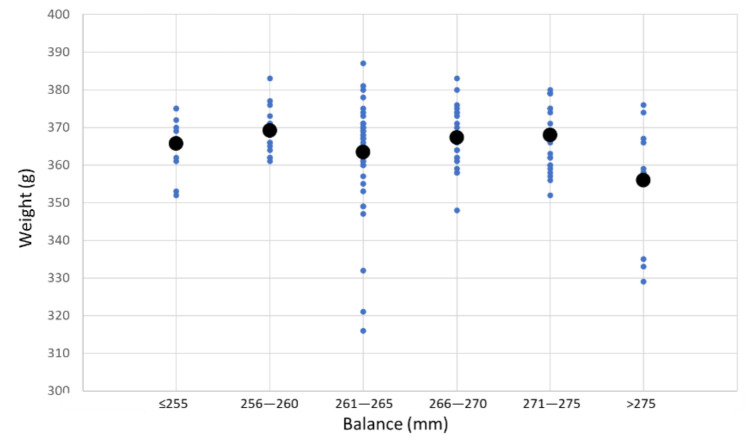
Weight, median values, and balance of 137 padel rackets.

**Figure 4 sensors-22-09266-f004:**
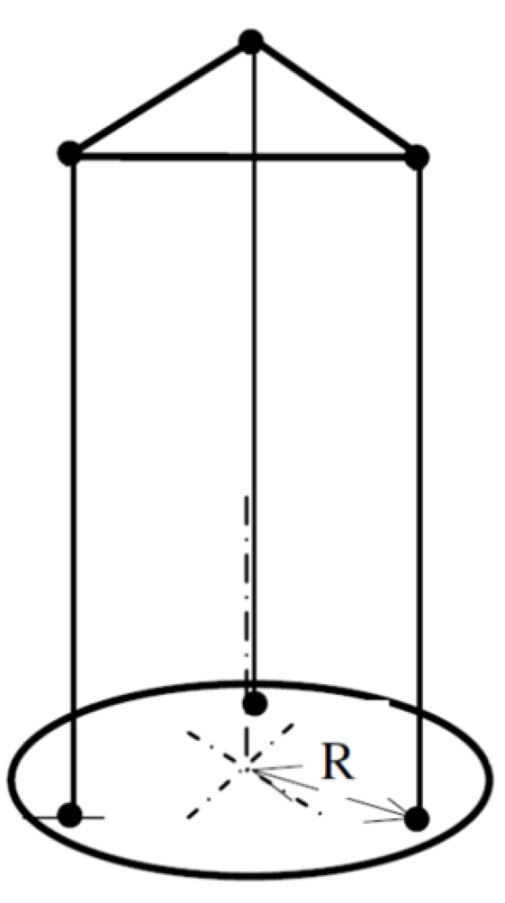
Trifilar pendulum.

**Figure 5 sensors-22-09266-f005:**
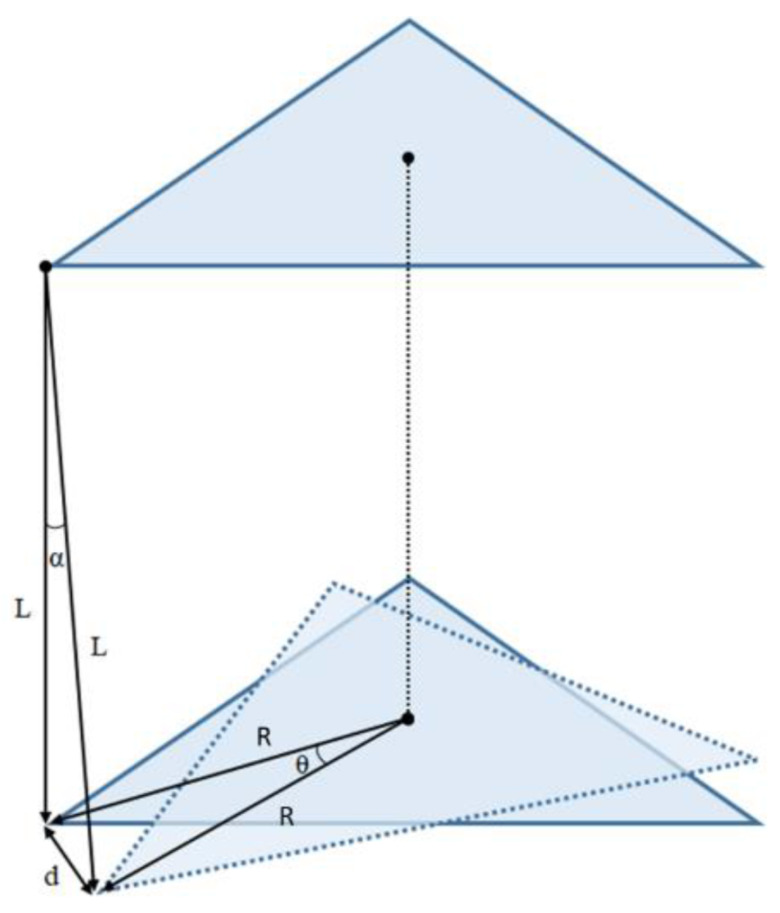
Displacement of the rotation body with labeled angles and lengths.

**Figure 6 sensors-22-09266-f006:**
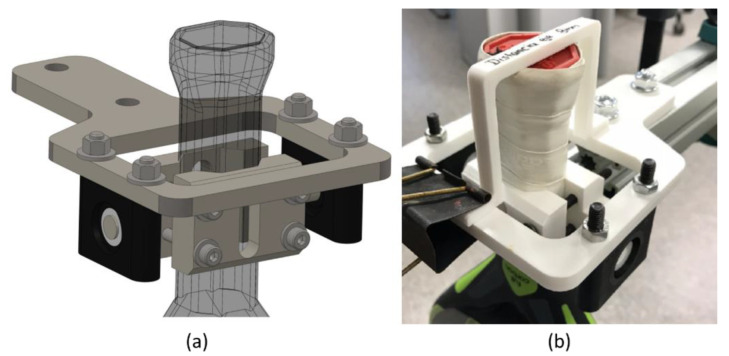
Setup for the horizontal rotation. (**a**) Computer-aided design (CAD) model. (**b**) The limiter positions the racket with a defined distance between the rotation axis and grip end.

**Figure 7 sensors-22-09266-f007:**
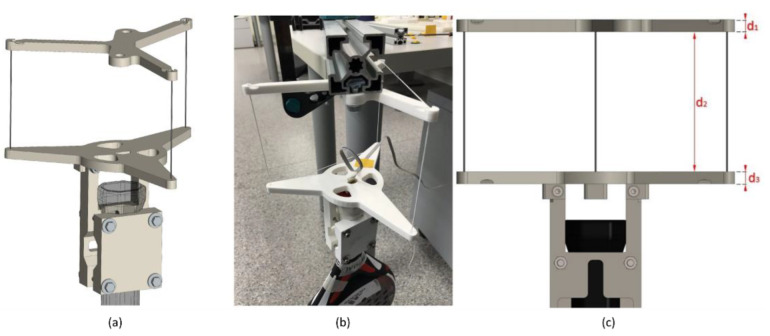
Setup for the vertical rotation. (**a**) CAD model, (**b**) final assembly, and (**c**) detail of strings.

**Figure 8 sensors-22-09266-f008:**
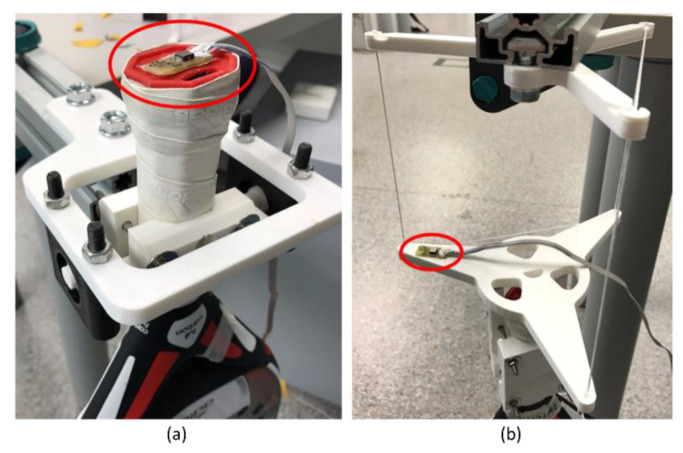
The location where the accelerometers are attached. (**a**) For horizontal pendulum and (**b**) for the vertical pendulum.

**Figure 9 sensors-22-09266-f009:**
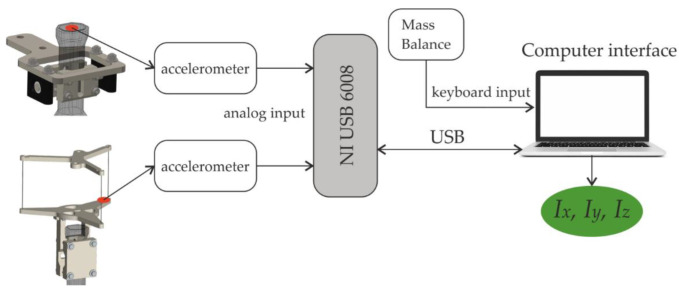
Experimental setup block diagram.

**Figure 10 sensors-22-09266-f010:**
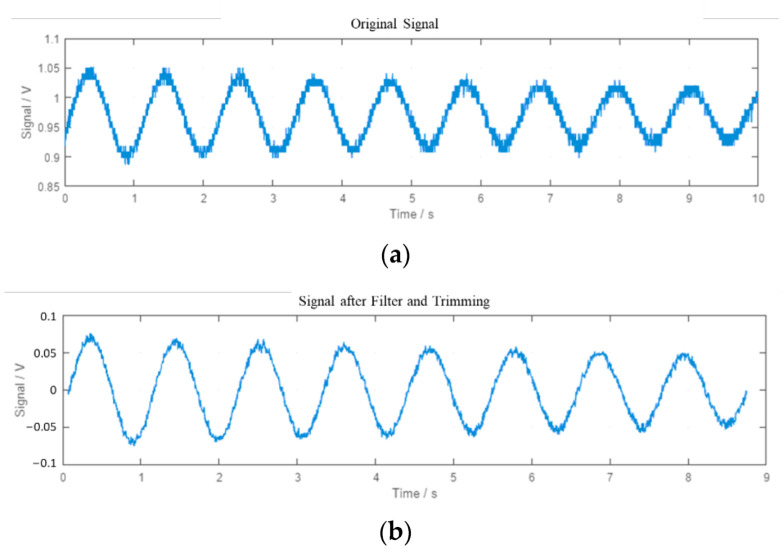
Plots of the signal in the different steps of the processing. (**a**) Raw signal. (**b**) Processed signal (filtered and trimmed).

**Figure 11 sensors-22-09266-f011:**
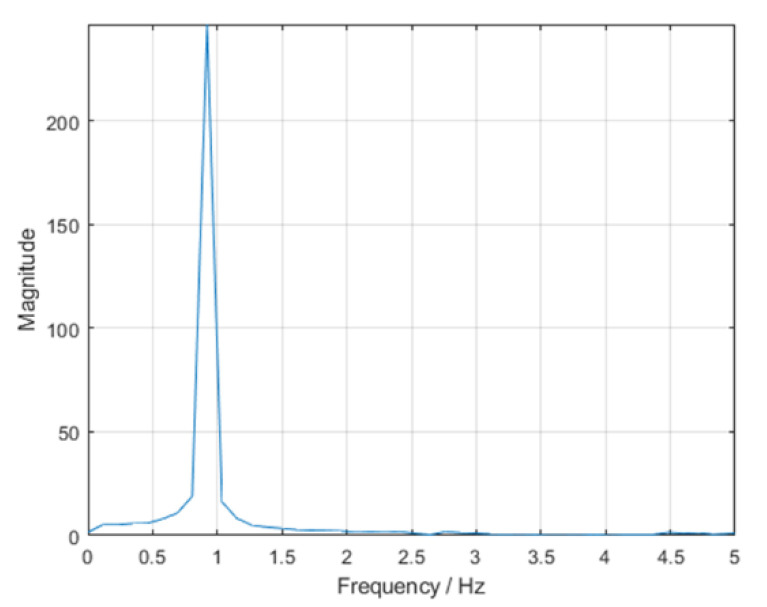
Frequency spectrum of the signal after the FFT. The dominating frequency can be found at 0.920 Hz, which corresponds to a period duration of *T* = 1.086 s.

**Figure 12 sensors-22-09266-f012:**
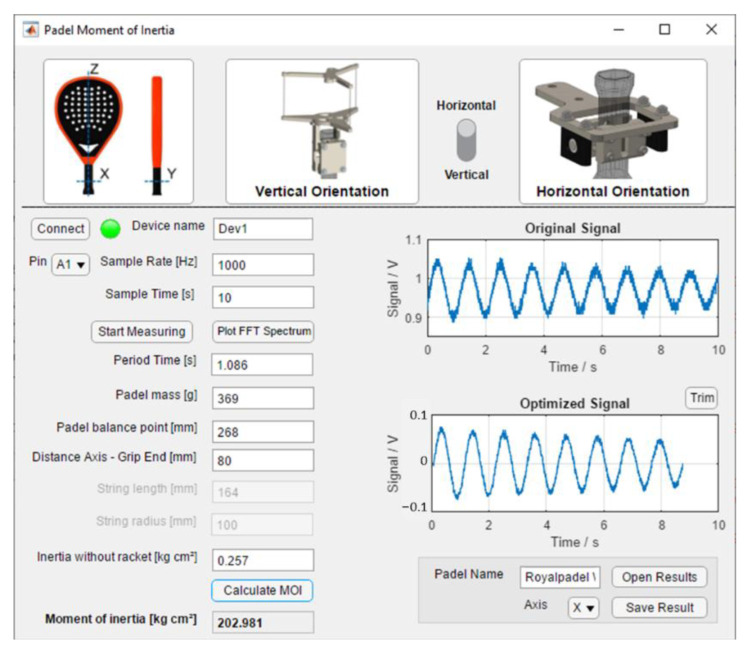
Screenshot of the MATLAB^®^ interface to run and store experiments.

**Figure 13 sensors-22-09266-f013:**
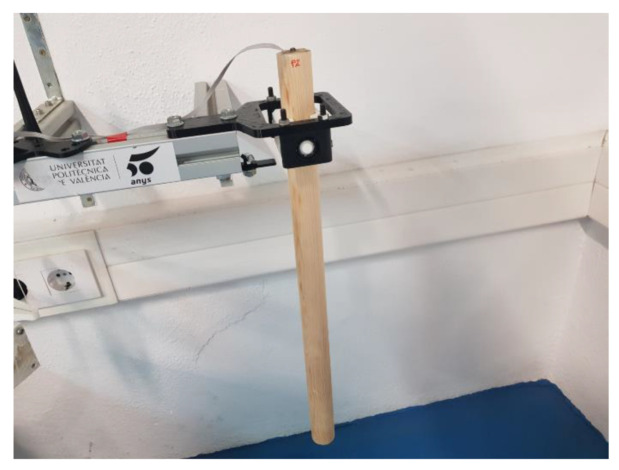
One of the references used to calibrate vertical MOI test stand.

**Figure 14 sensors-22-09266-f014:**
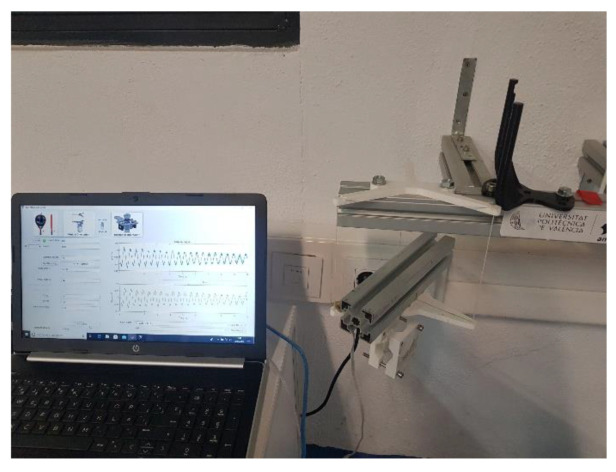
One of the references used to calibrate horizontal MOI test stand.

**Figure 15 sensors-22-09266-f015:**
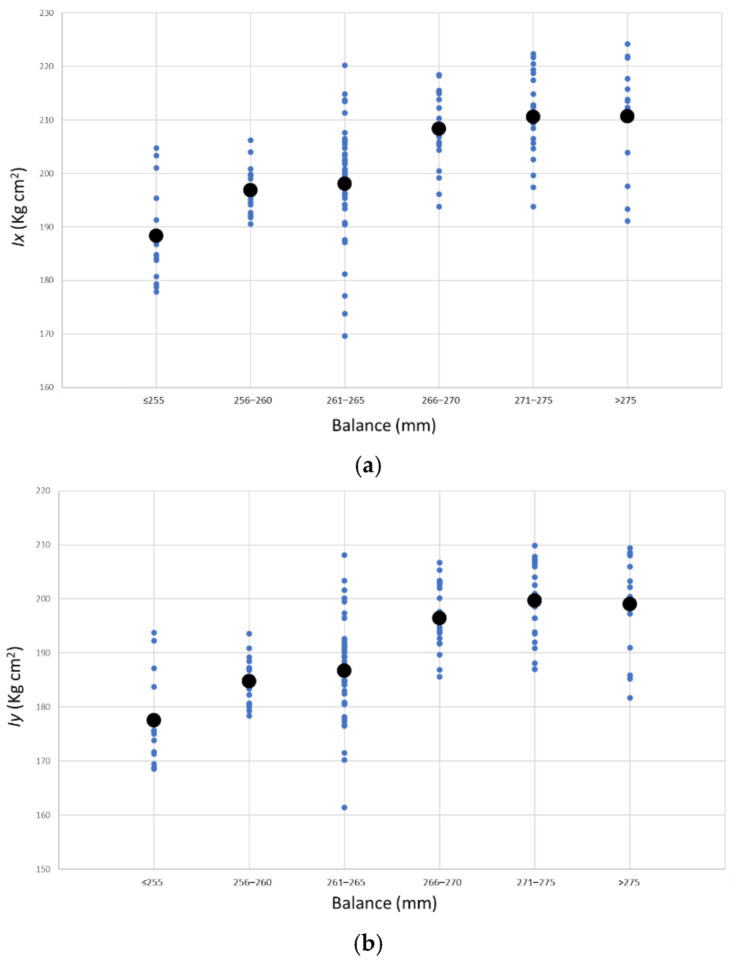
Moments of inertia *I_x_* (**a**) and *I_y_* (**b**) and median values as a function of the balance distance.

**Figure 16 sensors-22-09266-f016:**
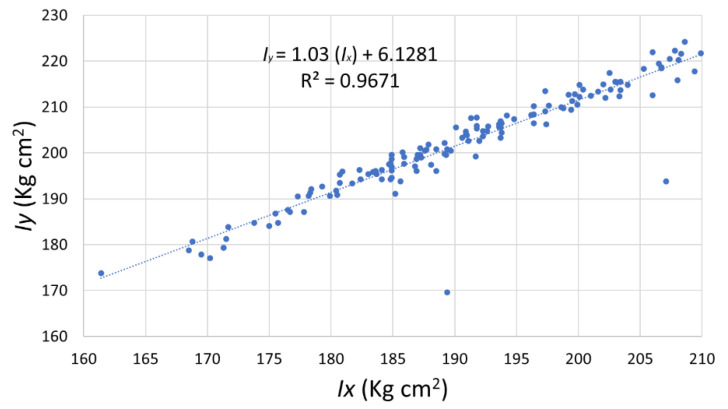
Moment of inertia *I_y_* as a function of the moment of inertia *I_x_*.

**Figure 17 sensors-22-09266-f017:**
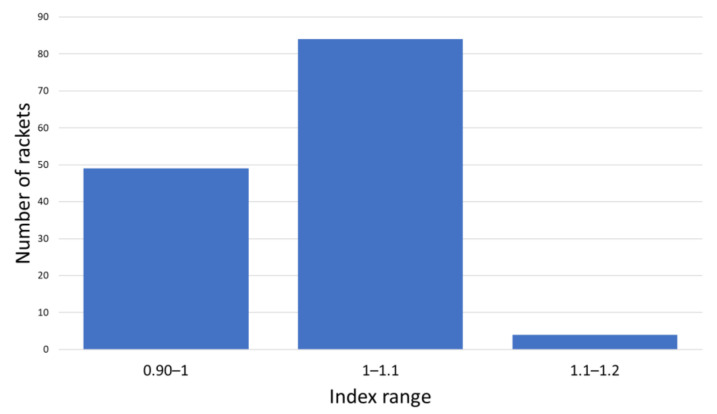
Results for the maneuverability parameter for the 137 padel rackets studied.

**Figure 18 sensors-22-09266-f018:**
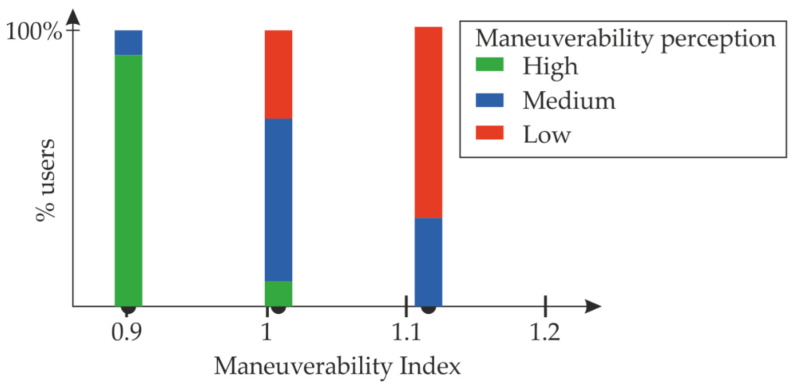
Results of the test with users.

**Table 1 sensors-22-09266-t001:** Comparison of the different approaches to determine the MOI experimentally.

Device	Advantages	Disadvantages
Atwood machine	Easy calculation	Long time for preparation Controllable motor recommended
Pendulum	Simple design	Different designs depending on the axis
Springs	Easy calculation Simple design	Spring characteristics must be determined

**Table 2 sensors-22-09266-t002:** Average values, partial derivation, standard deviation, and the influence of every parameter in error for the case of horizontal rotation.

Parameter	x_i_	∂I/∂x_i_	∆xi	∂I/∂x_i_∆xi10^−4^
*T*	1.103 s	0.0485	0.00121 s	0.587
*m_r_*	0.363 kg	0.0736	0.000516 kg	0.380
*d_r_*	0.265 m	0.1007	0.000425 m	0.428

**Table 3 sensors-22-09266-t003:** Results summary of the references used to calibrate vertical MOI test stand.

Model	Theoretical MOI X (kg cm^2^)	Measured MOI X (kg cm^2^)	MOI Standard Deviation	Accuracy Error (%)	Standard Deviation/Theoretical MOI (%)
a	202.936	203.69	0.704	−0.37	0.35
b	186.116	184.53	0.262	0.85	0.14
c	168.074	168.49	0.265	−0.25	0.16

**Table 4 sensors-22-09266-t004:** Average values, partial derivation, standard deviation, and the influence of every parameter in error for the case of vertical rotation.

Parameter	x_i_	∂I/∂x_i_	∆xi	∂I/∂x_i_∆xi10^−4^
*T*	0.4829 s	0.0084	0.00108 s	0.0903
*T* _0_	0.31008 s	−0.0016	0.0016 s	−0.0258
*m_r_*	0.3630 kg	0.0039	0.000516 kg	0.0201
*m* _0_	0.1568 kg	0.0023	0.000168 kg	0.0039
*L*	0.1498 m	−0.0018	0.000087 m	−0.0015
*R*	0.10008 m	0.0354	0.000174 m	0.0607

**Table 5 sensors-22-09266-t005:** Results summary of the references used to calibrate horizontal MOI test stand.

Model	Theoretical MOI X (kg cm^2^)	Measured MOI X (kg cm^2^)	MOI Standard Deviation	Accuracy Error (%)	Standard Deviation/Measured MOI (%)
A	20.000	19.750	0.0777	−1.14	0.39
B	17.506	17.527	0.0412	−0.12	0.24
C	15.231	14.971	0.0412	1.71	0.28

**Table 6 sensors-22-09266-t006:** Results for the relative deviation of experiments in the test stand.

Test Stand	Error Propagation (%)	Accuracy Error (%)	Repeatability (%)
Horizontal	0.418	0.491	0.215
Vertical	0.658	0.990	0.303

**Table 7 sensors-22-09266-t007:** Statistic values of the 137 padel rackets.

Parameter	*m* (g)	*d_com_* (mm)	*I_x_ (*kg cm^2^)	*I_y_* (kg cm^2^)	*I_z_* (kg cm^2^)
Avg	363.07	265.52	190.599	202.027	17.282
Min	316	246	161.405	169.631	14.633
Max	387	283	209.869	224.166	19.812

**Table 8 sensors-22-09266-t008:** Results of the test with users. Each value represents the percentage of users classifying the racket in the category.

Padel Racket Index	High	Medium	Low
0.90	91%	9%	0%
1.02	9%	59%	32%
1.12	0	32%	68%

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
