# Peer review of "Identifying the Inertial Properties of a Padel Racket: An Experimental Maneuverability Proposal"

_sensors, 2022, doi:10.3390/s22239266_

Round 1

Reviewer 1 Report

Summary:
The manuscript present a  solution to determine the moment of inertia around different axes of padel rackets and makes a standardized comparison possible. The authors use the approach based in the fact that a pendulum swings with its natural frequency, which depends directly on its moment of inertia and the inertia is calculated by measuring the cycle time of the swing. In addition, the precision is estimated by calculating the influence of the measurement errors and testing the repeatability. Software (application) with a graphical user interface is been programmed to perform the experiments; and the sensor data is acquired, visualized, and processed directly in the application, and the calculated results are stored in a database. In conclusion, authors mention that with the current solution and design, there is no easy way to increase the accuracy further; and if higher accuracy is required, another test stand with one of the other approaches can be developed to compare the precision of the results.

In the manuscript presented, the authors address an interesting experiments to determine the moment of inertia around different axes of padel rackets. In addiction, the manuscript is well organized and structured, nevertheless I consider that the authors should attend to certain minor issues in order to improve the good quality of work.

Comments and Suggestions for Authors:

- All Figures should be aligning center position.

- Even if the reader knows the subject, the authors should expand the first occurrence of any the acronyms or initialism, some of them are not expanded as examples CAD, PLA, etc.

- References to standards, de facto standard, services, equipment and components must be included, as example ADXL Dual-Axis accelerometer should be has reference to datasheet of specific part number.

- Authors should include a block diagram of the acquisition system (ADXL sensor, I/O device USB-6008, Personal Computer (PC), etc.) to provide an overview of it to the reader.

- Figure 6 (b) shows that the position limiter is fixed with a blinder clip. Is it just to fix the limiter plastic part before tightening the screws that hold the racket? please explain about it.

- Fig 9 should be updated using a higher quality image.

- Authors should use standard notation, on page 12 row 381 a percentage amount is shown as 0,490% (using a comma); it should be 0.490%.

.oOo.

Author Response

The authors would like to thank the anonymous reviewers for their valuable comments, which have greatly improved the quality of the paper.

Review in the attached file.

Reviewer 2 Report

The paper present a solution to determine the moment of inertia around different axes of padel rackets and makes a standardized comparison possible

The research idea is interesting but some considerations should be noted.

1. Highlight more the contributions of the work in the abstract and in the introduction.

2. Highlight in the abstract what is presented in the search results.

3. In the Introduction, a literature review should be carried out comparing other similar works. The work has few references.

4. Describe all terms of the equations presented.

5. In Section 3, detail better error estimation.

6. A discussion of the results should be presented, showing the advantages and disadvantages of the opposite method.

Author Response

(The authors gave the same response as above.)

Reviewer 3 Report

Please kindly see the attachment.

Author Response

(The authors gave the same response as above.)

Round 2

Reviewer 2 Report

This reviewer's requests were met, so the manuscript can be published.